# A Machine Learning-Based Anomaly Prediction Service for Software-Defined Networks

**DOI:** 10.3390/s22218434

**Published:** 2022-11-02

**Authors:** Zohaib Latif, Qasim Umer, Choonhwa Lee, Kashif Sharif, Fan Li, Sujit Biswas

**Affiliations:** 1Department of Computer Science, Hanyang University, Seoul 04763, Korea; 2Department of Computer Science, COMSATS University Islamabad, Vehari Campus, Vehari 61100, Pakistan; 3School of Computer Science and Technology, Beijing Institute of Technology, Beijing 100081, China; 4Computer Science and Digital Technologies Department, University of East London, London E16 2RD, UK

**Keywords:** software-defined networking (SDN), anomaly prediction, OpenFlow, machine learning (ML)

## Abstract

Software-defined networking (SDN) has gained tremendous growth and can be exploited in different network scenarios, from data centers to wide-area 5G networks. It shifts control logic from the devices to a centralized entity (programmable controller) for efficient traffic monitoring and flow management. A software-based controller enforces rules and policies on the requests sent by forwarding elements; however, it cannot detect anomalous patterns in the network traffic. Due to this, the controller may install the flow rules against the anomalies, reducing the overall network performance. These anomalies may indicate threats to the network and decrease its performance and security. Machine learning (ML) approaches can identify such traffic flow patterns and predict the systems’ impending threats. We propose an ML-based service to predict traffic anomalies for software-defined networks in this work. We first create a large dataset for network traffic by modeling a programmable data center with a signature-based intrusion-detection system. The feature vectors are pre-processed and are constructed against each flow request by the forwarding element. Then, we input the feature vector of each request to a machine learning classifier for training to predict anomalies. Finally, we use the holdout cross-validation technique to evaluate the proposed approach. The evaluation results specify that the proposed approach is highly accurate. In contrast to baseline approaches (random prediction and zero rule), the performance improvement of the proposed approach in average accuracy, precision, recall, and f-measure is (54.14%, 65.30%, 81.63%, and 73.70%) and (4.61%, 11.13%, 9.45%, and 10.29%), respectively.

## 1. Introduction

The global Internet comprises users that are usually connected through thousands of autonomous systems from different geographical regions. Managing such vast and intricate systems requires a broad range of services and management applications. However, implementing versatile applications is time-consuming and introduces complexity in traditional networking. Moreover, extracting important features for the analysis to detect traffic anomalies is complicated in traditional networks. In contrast, software-defined networking (SDN) is a paradigm that provides a clean separation between the control plane (central plane) and data plane (bottom plane) [1,2]. This separation transforms the physical network devices into simple forwarding elements. At the same time, the software-based controller becomes a decision-making entity for the underlying data plane, whereas the management plane (top plane) enforces different policies. A set of well-defined protocols or application programmable interfaces (APIs) [3] enable communication within these planes and among different software-defined networks. The routing decision is shifted to a centralized controller in SDN [4]; therefore, only programmable switches are used for routing. Although there are multiple protocols available in the literature (and industry), OpenFlow [5] is the most widely adopted protocol for SDN to enable communication between the control plane and the data plane. A programmable switch makes decisions based on the guided rules provided by the controller. For example, the switch matches the packet header with flow entries in the flow table when a new packet arrives and forwards the packet to the particular port if a rule is satisfied. Alternatively, the packet is either dropped or forwarded to the controller as a request to install a flow entry on the switch to transfer the packet [6]. Programmable networks have become an indispensable feature of most modern-day networks, including the Internet of things (IoT) and 5G networks [7]. The anomaly detection problem requires attention not only in computer networks but also in energy and power industries. Moreover, cyberattacks on the IoT [8], which is complex system of wireless sensors networks and traditional networks [9], require learning-based solutions.

The architectural constraints of SDN limit the forwarding elements to completely follow the control plane decisions [10]. Although this is the intended purpose, the localized network traffic analysis and decision making for anomalous behavior are ignored in the current controllers. Hence, the controller may install the flow rules against anomalous traffic. Anomalies can be analyzed and removed manually; however, this becomes impractical for dynamic environments. To this end, offline analysis implemented as flow rules for future traffic classification can be used, but these do not offer protection against new anomalies or automated run-time analysis and implementation.

Traffic anomalies may be categorized into different severity levels by using various tools, e.g., Snort. Anomalies of any severity levels should be identified before the controller installs flow entries. To avoid the installation against irregularities, a machine learning-based traffic anomaly prediction can improve the controller’s and SDN’s performance in general. Several approaches have been proposed for traffic-related activities, e.g., normal activities identification [11], intrusion detection [12], and anomaly detection [13], based on global network view, centralized control, dynamic flow installation, programmability, and software-based traffic analysis. Global network view and programmability help to analyze network traffic to find anomalies and react against the identified anomalies. Although different approaches perform traffic anomaly detection by using the network history data, to the best of our knowledge, none of them performs automatic traffic anomaly prediction to avoid anomalies from the live flow of traffic by using machine learning techniques to make the network secure.

In this work, we propose a machine learning-based approach for predicting traffic anomalies in SDN where a machine learning classifier is attached to the SDN controller. This classifier helps the SDN controller to provide information about the anomalies in the network traffic. In response, the SDN controller installs the flow rules against normal traffic, whereas it does not install flow rules for abnormal traffic. For this purpose, we first use Mininet [14] and a signature-based intrusion-detection system (i.e., Snort) [15] to build a large dataset with OpenFlow traffic, containing normal and abnormal requests. We preprocess the attributes (features) of requests, and then feature vectors are constructed against each flow request. Subsequently, we input the feature vector of each anomaly into a machine learning classifier for training. Finally, the holdout cross-validation technique is adopted to evaluate the proposed approach. The major abbreviations used in this paper are presented in Table 1.

The main contributions of this work are as follows.
The machine learning-based approach is proposed for an automatic anomaly prediction for SDN (annotated as MAP-SDN).MAP-SDN helps the SDN controller to identify the network anomalies.It helps the SDN controller to install the flow rules for normal traffic; however, no flow rules are installed for abnormal traffic.The evaluation results specify that MAP-SDN is accurate and its average accuracy, precision, recall, and f-measure are 95.27%, 98.70%, 98.45%, and 98.57%, respectively.

The rest of the paper is organized as follows: The background information and related works in machine learning anomaly detection are presented in Section 2. The proposed approach of MAP-SDN is introduced in Section 3. The procedure to evaluate and discussion on results is given in Section 4. Section 5 concludes the paper and suggests future directions.

## 2. Background and Related Works

In this section, we first briefly introduce the baseline technologies and architectures, followed by the existing research works in anomaly detection for programmable networks.

### 2.1. Intrusion and Anomaly Detection in SDN

SDN paradigm provides a centralized and unified control by breaking the network into the control plane and data plane, as illustrated in Figure 1. In the multi-layered architecture of SDN, devices are managed and programmed by the controller in the control plane. Notably, southbound interfaces (protocols) enable communication between the control and data planes. A management plane enforces various network policies and programs. Some of the base functions of an SDN controller are also shown in Figure 1. The topology manager maintains the topology information based on the devices’ link information. The device manager keeps the information of underlying elements with the help of Packet_IN requests and identifies the uniqueness of devices by using MAC address and VLAN. The routing and forwarding function is responsible for installing flow rules on forwarding elements. The proposed module integrated with base network functions to mitigate anomalous flow installation from abnormal traffic requests is that MAP-SDN is the proposed module integrated with base network functions.

Network intrusion-detection systems (NIDS) are usually adopted to detect anomalies in traffic to determine if unwanted (malicious) packets are flowing into the network [16]. NIDS can be classified into two types; anomaly-based detection (e.g., PHAD [17]) and signature-based detection systems (e.g., Snort [15]). Anomaly-based detection systems look at network traffic to detect incorrect or abnormal activities. The major drawback of anomaly-based detection systems is that these systems cannot analyze custom protocols. In contrast, signature-based detection systems follow a proactive approach, use pattern-matching techniques, and analyze custom protocols. Signature-based detection systems are very simple to implement and have high accuracy and low false-positive rates compared to anomaly-based detection systems.

Anomaly detection in SDN has been mostly done through additional modules at the controller or specialized applications in the management plane. RAD [18] is an external application that runs in a management plane and provides an agile system for anomaly detection in SDN. It has several modules for traffic capturing and rule generation; however, the anomalies are detected by using Snort. Similar to RAD, Luiz et al. [19] proposed an ecosystem that tries to detect and mitigate anomalies in real time. It uses a traffic statistic collection module to monitor network traffic and extract information which includes source and destination IPs, source and destination ports, packets, and bits. In the anomaly-detection module, it uses two monitoring phases. In the first phase, it observes and compares traffic behavior to expected or normal behavior. If it finds any deviation, ostensible monitoring is launched in the second phase to identify the anomalies. In the case of anomalies, a mitigation module may perform several actions, including drop packet, change packet routing, and exclude anomalous flow entries. Both solutions are based on comparing the traffic to an existing traffic profile. Hence, for high-diversity traffic, the efficiency of such systems drops. Moreover, there is no earning process, so new anomalies cannot be detected.

### 2.2. Anomaly Detection in SDN by Using Machine and Deep Learning

The use of machine learning in SDN has been studied by Xie et al. [20] for various purposes. Nanda et al. [21] present a comparison between four machine learning algorithms (i.e., C4.5, Decision Table (DT), BayesNet (BN), and naive Bayes (NB)) for detecting network attacks. Their results indicate that BN performs better than the rest of the algorithms. Wang et al. [11] proposed a behavior-based SVM to categorize normal and intrusion traffic. In [22], the authors present a model that uses signature-based Snort IDS to detect anomalies in the SDN environment. A threat-aware system for intrusion detection is proposed in [23], which has three major subsystems: data preprocessing, predictive data modeling, and a decision-making and response subsystem. The data preprocessing subsystem is used to extract and select appropriate features. The predictive data modeling subsystem implements decision tree and random forest algorithms to predict intrusions. In contrast, the decision-making and response subsystem is used to install flow rules for different types of flows.

Hurley et al. [24] proposed another NIDS by using hidden Markov models [25]. HMM are trained by using the Baum–Welch algorithm and use source and destination IP and port and length of the packet as selected features. ATLANTIC [26] is another approach for detecting, classifying, and mitigating some anomalies. It uses information theory to calculate deviations in the entropy of flow tables and a machine learning algorithm based on SVM to analyze and classify flows according to their abnormal behavior.

Aleroud et al. [27] categorized anomalous events into three groups: attacks on the SDN control plane, compromising data and control plane communication, and threats for data plane elements. To detect distributed denial of service (DDoS) attacks, Barki et al. [28] proposed a new IDS in an SDN controller which uses the signature-based IDS that uses various algorithms (e.g., naive Bayes, KNN, K-means, and K-medoids) to classify normal and abnormal traffic. Similarly, work in [12] discusses various machine learning techniques for DDoS and intrusion prevention in SDN and provides a comparison between these techniques. Similar attacks are detected with the help of machine learning techniques in [29] where authors use SVM, NB, KNN, RF, and LR.

Although the works mentioned above use machine learning algorithms for anomaly detection, most of them only detect the attack pattern. Moreover, few approaches use IDS as part of the system. The proposed work differs from them as it avoids third-party tools and suggests an automatic approach by which to detect abnormal traffic to secure SDN.

Deep learning techniques have also been applied to SDN recently for feature learning [30]. Tang et al. [31] propose an IDS based on a gated recurrent unit recurrent neural network (GRU-RNN) with 89% accuracy with six raw features of flow statistics. In [32], authors exploit an autoencoder and LSTM-based deep learning approach to handling flow-based DDoS attacks in SDN. Dey et al. [33] used gated recurrent unit long short-term memory (GRU-LSTM) for the flow-based anomaly detection and implemented ANOVA F-test, recursive feature elimination, and feature selection methods. The results show an accuracy of 87% with the NSL KDD dataset with a very low false alarm rate which is 0.76%. Dawoud et al. [34] performed anomaly detection by adding an IDS module in the SDN controller and using TensorFlow as a deep learning library. Tang et al. [35] proposed a network intrusion-detection system implemented in the controller by using network status information. It shows that by reducing the learning rate, the loss reduces and accuracy increases. These deep learning solutions also incorporate IDS at different levels and require significant improvement to work efficiently in real time. The proposed approach differs from them as it does not require an IDS as part of the running system. Graph-based deep learning is another emerging technology that is applied in wired and wireless communication networks [36]. In [37], authors propose a new hierarchical adversarial attack (HAA) mechanism by using the graph neural network (GNN). The proposed approach can examine the generality and robustness of NIDS for IoT applications. Similarly, authors in [38] propose E-GraphSAGE, which is also a GNN-based approach. It can capture the edge features of the graph and topological information for NIDS in IoT networks.

## 3. Proposed Approach (MAP-SDN)

This section presents the details of the machine learning-based anomaly prediction for the SDN (MAP-SDN) scheme. The proposed scheme identifies the traffic anomalies for SDN and classifies them into two fundamental categories: normal or abnormal. It allows the SDN controller to install flow entries for normal traffic, whereas it avoids abnormal traffic. It is important to note here that abnormal traffic can be further classified into different severity levels, and accordingly, different policy actions can be taken. For simplicity, we only use two classification levels in the following mathematical representations.

Let a packet *p* from a set of network packets *P* be represented as
(1)p=<r,s>,
where r represents the set of attributes of *p*, and *s* is an assigned priority to *p*. In SDN, the first packet of a new flow trims a Packet_In message against which a flow is installed [6]. Here, *p* is such a packet that will trigger a Packet_In message from the switch to the controller.

Here, MAP-SDN performs anomaly classification of a new packet *p* either as *normal* or *abnormal*. This classification into category *c* can be represented as a function *f*, such that,
(2)$$c=f\left(p\right)$$
(3)$$c\;\in \left\{normal,abnormal\right\},\;p\;\in \;P,$$
where *c* is the classification result (i.e., normal or abnormal), *f* is a categorizing function, *p* is an input packet of the function, and *P* is a set of packets.

For anomaly prediction, we generate traffic on an edge-core network to collect the (PCAP) files generated by each network element. Although the overview of MAP-SDN workflow is shown in Figure 2. Algorithm 1 presents the process of anomaly prediction that takes PCAP as an input and returns a machine learning-based classifier for network traffic anomaly prediction.
**Algorithm 1** Network Traffic Anomaly Prediction1:**procedure**Anomaly_Prediction(PCAP)2:      Read *PCAP* files3:      Pass *PCAP* files to Snort for their structural information as an output (*alert_full*) for each *PCAP* file4:      **for** each *alert_full i* in PCAP **do**5:         Preprocess *i* to extract packets’ information, i.e., *source & destination IPs*, *source & destination ports*, and *protocol type*6:         Consturct vector for each *PCAP* file against its preprocessed information7:      **end for**8:      Given the constucted vectors, train a Machine Learning-based Classifier (MLC)9:      Return MLC10:**end procedure**

The following sections explain each of the key steps of MAP-SDN.

### 3.1. Data Acquisition

To construct the dataset of traffic anomalies, we design an edge-core-based topology as shown in Figure 3. It is important to note that the available dataset is not enough to train the classifier. We have used the datasets from [39]; however, most datasets have an uneven distribution of normal and abnormal traffic. Therefore, we have augmented this dataset with a synthetic dataset by using the following process. The topology (shown in Figure 3) consists of backbone (core) part and edge part, where switches S1–S4 act as backbone and S5–S12 act as edge switches which provide connection to hosts H1–H16.

In order to generate a variety of traffic patterns, we use *Iperf* [40] and *hping3* [41] and obtain the *PCAP* files from each of the edge elements. Then, we use Snort to extract data from these files based on a pre-defined rule database for data extraction. The major reason to use Snort is to obtain the severity of the abnormal traffic. We input PCAP to Snort to generate full packet headers by using the alert_full configuration plugin. Although Snort has many pre-defined rules, it also supports user-defined rules. We define our own rule to extract data from our network topology. An example defined rule is given as
$$alert\;P\_type\;S_{ip}\;S_{port}\to D_{ip}\;D_{port}$$
$$(M_{text};\;ID;\;C_{type};\;R;),$$
where alert is a rule action which generates an alert when the set condition is met, $P_{type}$ is a required protocol (*TCP*, *UDP*, or *ICMP*) on which Snort generates alerts, $S_{ip}$ is a source IP of *i*th host, $S_{port}$ is a source port, arrow (→) is a representation of direction from source to destination, $D_{ip}$ is a destination IP of *i*th host, $D_{port}$ is a destination port, $M_{text}$ represents a message with Snort alert, ID represents Snort rule ID, $C_{type}$ is a predefined Snort category which helps with rule organization, and *R* is the default priority of the classification that can be modified by using a priority keyword inside the rule options. Notably, the use of any keyword for $S_{ip}$, $D_{ip}$, $S_{port}$ and $D_{port}$ generates alerts from any IP or port. The output of Snort is a structured text file against the given rule. This output is shown in Figure 4. It is worth mentioning that the main objective of this paper is to enable the SDN controller to classify normal and abnormal traffic. In addition, it helps the SDN controller not to install flow rules against abnormal traffic.

### 3.2. Preprocessing

The Snort output contains some repetitive parameters, e.g., *message_text*, *class_type*, etc. as shown in Figure 4. Such repetitive parameters of the output are overhead for feature modeling. Therefore, we pre-process the structured file to extract the useful attributes only. This is achieved by using a custom Python script that separates each attribute of each output file. It extracts the required parameters, i.e., (class_type, priority, time, source IP, and destination IP, source port, destination port, protocol_type, datagram length, time to live (TTL), IP length, and datagram length), avoids the repetitive parametric values, and stores them.

After preprocessing, a packet *p* is represented as
(4)$$p=<r^{\prime },s>$$
(5)$$r^{\prime }=<a_{1},a_{2},a_{3},\cdots ,a_{l}>,$$
where $r^{\prime }$ and l is a set of preprocessed selected attributes $a_{1}$, $a_{2}$, $a_{3}$, …, $a_{l}$ of each packet from the Snort output, and length of attributes, respectively. Note that $a_{i}$ is an 〈attribute, value〉 pair.

### 3.3. Feature Modeling

We create a high-dimensional feature matrix where each packet represents a row and $a_{1}$, $a_{2}$, $a_{3}$, ⋯, $a_{n}$ represent the columns of the matrix, respectively. A feature vector of a packet *p* can be formalized as
(6)$$p=<f_{1},f_{2},f_{3},\ldots \ldots ,f_{l})>,$$
where $f_{1},f_{2},f_{3},\ldots \ldots ,f_{l}$ and *l* represent the feature set of each packet and length of features, respectively.

To populate the feature matrix, we define a rule to assign the feature values to $f_{i}$. The rule assigns the $v_{i}$ value (extracted from $a_{i}$) to $f_{i}$ if found, otherwise marked 0. The conditions used to mark the values to the feature can be represented as
$$f_{i}\left(p\right)=\left\{\begin{array}{c} 0,\;if\;v_{i}\notin r^{\prime }\\ v,\;if\;v_{i}\in r^{\prime }, \end{array}\right.$$
where **$f_{i}$** represents the feature set of each *p*.

### 3.4. Training of the Model

MAP-SDN uses the random forest (RF) classification algorithm (which is a tree-based algorithm) for the prediction of traffic anomalies. In this model, several (decision) trees are built, and the output of the trees is aggregated to increase the generalizability. This process is an ensemble method that combines weak learners (i.e., individual trees) to produce a strong learner [42]. Here, the RF classifier is defined as a collection of tree-structured classifiers g(x,θk),k=1,…, where θk represents independent, identically distributed random vectors (i.i.d) and every tree outputs a single vote for the most popular class at input. Initially, *n* packets are randomly picked from the dataset, and decision trees are constrained accordingly. Following this, every tree individually predicts traffic anomaly. Finally, a *c* is suggested to each new packet *p* considering the majority votes of *c* collected from the decision trees.

Let $P=p_{i},p_{2},\cdots ,p_{n}$ represents the training dataset having *n* training samples. After the preprocessing and feature modeling of *P*, each $p_{i}$ has a list of attributes ($f_{1},f_{2},\ldots ,f_{m}$), where *m* is the length of the features. A decision is constructed for each training sample. The model selects the most significant attributes to build the decision trees. Note that RF randomly selects the attributes where the selected attributes should be less than *m*. To this end, it uses *gini* index that can be presented as,
(7)$$gini\left(f_{j}\right)=1-\sum {\left[F_{j}\right]}^{2}$$
(8)$$gini_{split}=\sum_{f=1}^{m}\frac{n_{p}}{n}gini\left(f_{j}\right),$$
where, $F_{j}$ and $n_{p}$ are the relative frequency of *j*th attribute $f_{j}$ and randomly selected training samples among total training samples *n*. $gini_{split}$ constructs the decision trees and leads all of them as the tree’s root node.

## 4. Evaluation and Analysis

In this section, we first develop the research questions needed to be evaluated to establish the efficiency of MAP-SDN. Following this, we describe the evaluation criteria and the collected results with their analysis. Finally, we present the threat to validation for the proposed scheme to ensure the repeatability of the proposed approach.

### 4.1. Research Questions

The evaluation investigates the following research questions:RQ1: How accurate is MAP-SDN in anomaly prediction for SDN?RQ2: Does RF surpass other off-the-shelf algorithms?RQ3: Does pre-processing influence the performance of MAP-SDN? If yes, to what extent?

The RQ1 is constructed to investigate the accuracy of MAP-SDN. It compares MAP-SDN with two baseline prediction algorithms, i.e., the random prediction algorithm (RPA) and the zero rule algorithm (ZRA). These algorithms are considered a baseline approach in the literature when working on a unique problem. We also set up an environment by using a simulation tool to evaluate the performance of MAP-SDN.

The RQ2 compares the performances of different machine learning algorithms. This comparison reveals whether RF outclasses the off-the-shelf algorithms in predicting traffic anomalies.

The RQ3 investigates the impact of preprocessing by comparing the performance of MAP-SDN with and without preprocessing of the dataset.

### 4.2. Dataset and Metrics

Using the topology and data generation steps described earlier, different hosts of Figure 2 use *Iperf* and *hping3* to generate realistic traffic patterns. The generated dataset contains 101,336 samples in which 70.28% are normal activities and 27.71% are abnormal activates. Note that we use the realistic traffic only to create the dataset instead of passing realistic traffic to the proposed approach. Because we used the proposed approach on sample data, which is not very extensive, we did not find any significant overhead of time consumption or time complexity. However, we can explore the time complexity in the future.

The evaluation metrics used to evaluate MAP-SDN in this work are accuracy, precision, recall, and f-measure. These matrices have been extensively used in the literature and are recommended for machine learning classification problems [43,44,45]. The formulas used to calculate each one are given below:$$accuracy=\frac{TP+TN}{TP\;+\;TN\;+\;FP\;+\;FN}\;,$$
$$precision=\frac{TP}{TP\;+\;FP}\;,$$
$$recall=\frac{TP}{TP\;+\;FN}\;,\;\mathrm{and}$$
$$f-measure=\frac{2\;\times \;precision\;\times \;recall}{pecision\;+\;recall}\;.$$Here, TP are the truly predicted samples as normal from $P_{tr}$, TN are the truly predicted samples as abnormal from $P_{tr}$, FP are the incorrectly predicted samples as normal from $P_{tr}$, and FN are the incorrectly predicted samples as abnormal from $P_{tr}$.

### 4.3. Evaluation Process

To evaluate MAP-SDN, we first exploit the PCAP file by using edge-core-based topology and Snort to extract the data from network. Then, we pre-process the extracted data to extract the required attribute mentioned in Section 3.2. Next, hold-out validation is performed on *P* that splits P into 80–20%. We use 80% of all packets as training set $P_{tr}$, whereas 20% of remaining packets are used as testing set $P_{te}$. Notably, we split $P_{te}$ into *i*th combinations notated as $m_{i}\left(i=1,\ldots ,4\right)$ where $m_{1}$, $m_{2}$, $m_{3}$, and $m_{4}$ contain 5%, 10%, 15%, and 20% packets out of total $p_{te}$, respectively.

For each $p_{i}$, the following process is applied.
We select the training set $P_{tr}$ and train the naive Bayes (NB), multinomial naive Bayes (MNB), linear regression classifier (LR), random forest classifier (RF), support vector machine (SVM), and decision tree (DT) classifiers on $P_{tr}$.Then, for each *i*th sample from testing set $P_{te}$, we predict the traffic anomalies by using trained classifiers (NB, MNB, LR, RF, SVM, and DT, respectively).Finally, we calculate and compare the performances of all classifiers by using the evaluation metrics, i.e., accuracy, precision, recall, and f-measure.

### 4.4. Analysis of Results

#### 4.4.1. RQ1—Accuracy of MAP-SDN

We compare the proposed MAP-SDN with two baseline algorithms (RPA and ZRA) to investigate RQ1. Notably, these are the baseline approaches to verify the accuracy of MAP-SDN. The reason for choosing these algorithms as benchmarks has already been discussed in Section 4.1.

The evaluation results of MAP-SDN, RPA, and ZRA are presented in Table 2. The testing iterations *i* are presented in the first column, followed by the accuracy, precision, recall, and f-measure of each classifier individually. The last row of Table 2 presents the average results of the classifiers. We present the f-measure distribution of hold-out cross-validation of MAP-SDN, RPA, and ZRA in Figure 5. It contains a bean of each approach for f-measure results, where each bean contains small horizontal lines against *i* cross-validations and a long horizontal line against the average of cross-validations.

From the Table 2 and Figure 5, we notice the following:The MAP-SDN performs better than the RPA and ZRA in accuracy, precision, recall, and f-measure.In contrast to RPA, the performance improvement of MAP-SDN in accuracy, precision, recall, and f-measure is 54.14% = (95.27% − 61.81%)/61.81%, 65.30% = (98.70% − 59.71%)/59.71%, 81.63% = (98.45% − 54.20%)/54.20%, and 73.70% = (98.57% − 56.75%)/56.75%, respectively.In contrast to ZRA, the performance improvement of MAP-SDN in accuracy, precision, recall, and f-measure is 4.61% = (95.27% − 91.07%)/91.07%, 11.13% = (98.70% − 88.82%)/88.82%, 9.45% = (98.45% − 89.95%)/89.95%, and 10.29% = (98.57% − 89.37%)/89.37%, respectively.The average performance of MAP-SDN is better than the highest performances of RPA and ZRA as shown in Figure 5.

Moreover, we notice that MAP-SDN computes a few false positives and false negatives. The reason for this misclassification could be the use of Snort to construct the dataset. However, in the future, we will investigate the details to figure out the measures to reduce misclassification.

In order to further analyze the performance of MAP-SDN, we determine the significant differences between MAP-SDN, RPA, and ZRA. To this end, we first perform a one-way analysis of variance (ANOVA) and then confirm the result of ANOVA by applying the Wilcoxon test. Notably, we perform ANOVA and Wilcoxon tests with default settings in Excel and Stata, respectively. The f-ratio value and *p*-value of ANOVA are 45.16 and 1.79 × 10${}^{-5}$, whereas the *p*-value of Wilcoxon test is 1.57 × 10${}^{-3}$. Both ANOVA and Wilcoxon test confirms that the factor (using different approaches) has a significant difference at p<0.05.

Real-time Accuracy of MAP-SDN: We perform an experiment to check the real-time accuracy of MAP-SDN. We exploit the Mininet simulation tool to setup our topology (shown in Figure 3). In the topology, H1 forward anomaly traffic to victim host H5 and at the same time, host H2 sends normal traffic to host H9. To compare the results of MAP-SDN, we use Floodlight [46] in control plane as SDN controller and alternatively include Snort and MAP-SDN as anomaly detection tool. The MAP-SDN is working as an application of Floodlight controller. The Floodlight controller interacts with the MAP-SDN before installing the flow rules.

We develop an external application to extract the controller’s device-level information, build a global view of the whole network, and mark all possible paths from source to destination. Several pre-defined rules are placed in Snort database as a signature to detect anomalies. Based on these rules, Snort decides whether traffic is suspicious or not. To mitigate the suspicious traffic from the network, Snort matches source IP, destination IP, source port, and destination port with pre-defined rules. This matching results in a priority-based output which is further used by the application. In the case of anomalies, the application installs a flow on the source switch to drop all the packets from the attacker host. On the other hand, we replace the application and Snort with MAP-SDN to check the accuracy of the proposed approach. The results of both scenarios are shown in Figure 6.

From the Figure 6a,b, we notice the following:MAP-SDN has a significant difference in performance against application using Snort in bandwidth and data transfer scenarios.The detection and mitigation time in both (bandwidth and data transfer) is very high. Figure 6b represents data transferred that increases from 20 GB to 35 GB after mitigating anomalies. Similarly, Figure 6a represents bandwidth that significantly increases from 2.5 Gbps to more than 4 Gbps. The jump in both figures at the 40-s mark is due to the normal flow rules.MAP-SDN detects the anomalies early and significantly improves the bandwidth utilization and data transfer rate by avoiding flow installation against anomalous traffic.

The initial analysis concludes that MAP-SDN accurately predicts SDN traffic anomalies.

#### 4.4.2. RQ2—Performance Comparison of Off-the-Shelf Algorithms

We leverage widely adopted classification algorithms (MNB, LR, RF, and SVM) due to their competitive performance [43,44,45,47] to investigate RQ2.

The evaluation results of MNB, LR, RF, and SVM are presented in Table 3. The accuracy, precision, recall, and f-measure of each classifier are presented in the columns 2–5, respectively. From Table 3, we notice the following:RF yields the most accurate results. RF outperforms MNB, LR, and SVM in accuracy, precision, recall, and f-measure, respectively. The reason is that RF achieves better results due to its degree of freedom.MNB surpasses LR and SVM and its performance is very close to the proposed classifier RF.

The preceding analysis concludes that the results of MAP-SDN are significantly better with RF classifier.

#### 4.4.3. RQ3—Influence of Preprocessing

To investigate RQ3, we compare the results of MAP-SDN by enabling and disabling preprocessing. The evaluation results of MAP-SDN on different settings for preprocessing are presented in Table 4. The accuracy, precision, recall, and f-measure of MAP-SDN are presented in the columns 2–5 of table, respectively. The performance of MAP-SDN on different settings for preprocessing is presented in the rows of Table 4, respectively. The last row presents the improvement percentage in performance with preprocessing.

From Table 4, we make the following observations:MAP-SDN performs significantly better when preprocessing is used. The results show that the performance improvement in accuracy, precision, recall, and f-measure is 2.07% = (95.27% − 93.34%)/93.34%, 3.20% = (98.70% − 95.64%)/95.64%, 3.95% = (98.45% − 94.70%)/94.70%, and 3.57% = (98.57% − 95.17%)/95.17%, respectively.Without preprocessing, the performance of MAP-SDN is significantly impacted. One possible reason for the decrease in performance is that, without preprocessing, the model may include unwanted features.

The preceding analysis concludes that the preprocessing of the SDN traffic packets is an essential step for MAP-SDN.

### 4.5. Threats to Validity

The chosen metrics (accuracy, precision, recall, and f-measure) for evaluating MAP-SDN could be a threat to construct validity. The reason to choose these metrics is their popularity and performance for the machine learning classification problems [43,44,45].

The leverage of Snort to identify the anomalies in SDN traffic could be another threat to validity. To the best of our knowledge, Snort is the only tool that semi-automatically identifies traffic anomalies. The usage of other tools could affect the performance of MAP-SDN.

Another threat to construct validity is related to the generated dataset. The dataset with additional parameters (generated by other IDS tools) or the usage of other available datasets could improve the performance of MAP-SDN.

The abstraction of MAP-SDN could be a threat to external validity. We designed our topology and generated the dataset by using different tools. The generalized dataset for SDN traffic may affect the performance of MAP-SDN.

Another threat to external validity is a small dataset. Therefore, we use traditional machine learning algorithms to evaluate the proposed approach. Another reason for selecting a machine learning classifier is that deep learning algorithms mostly require significant training data.

## 5. Conclusions

The anomalies in SDN traffic are critical for the efficiency and security of programmable networks. Automatic identification of traffic anomalies in SDN could improve the performance and protect the network, the control plane, and the end hosts. To this end, in this paper, we proposed a machine learning-based approach for predicting traffic anomalies in SDN. The proposed approach preprocesses the data samples from a specially generated dataset and a signature-based intrusion-detection system. A feature vector is constructed against each sample, and subsequently, based on these feature vectors, a machine learning classifier is trained to predict traffic anomalies. Finally, the hold-out validation technique is utilized to evaluate the proposed approach. The evaluation results indicate that the proposed approach (PCAP) does not accurate against the baseline approaches (zero rule algorithm and random prediction algorithm), but also outperforms the other well-known machine learning algorithms (linear regression, support vector machine, and multinomial naive Bayes) for classification.

In the future, we will extend this work to identify the minor false positive, and false negative results observed, as well as improve the dataset along with the classification process. It will also be interesting to expand the anomaly classification to a higher granularity and allow more control policies accordingly. It would be interesting to find the impact of higher granularity and allow more control policies for network traffic anomaly prediction. Exploring such features from an average dataset length would help deep learning approaches improve performance.

## Figures and Tables

**Figure 1 sensors-22-08434-f001:**
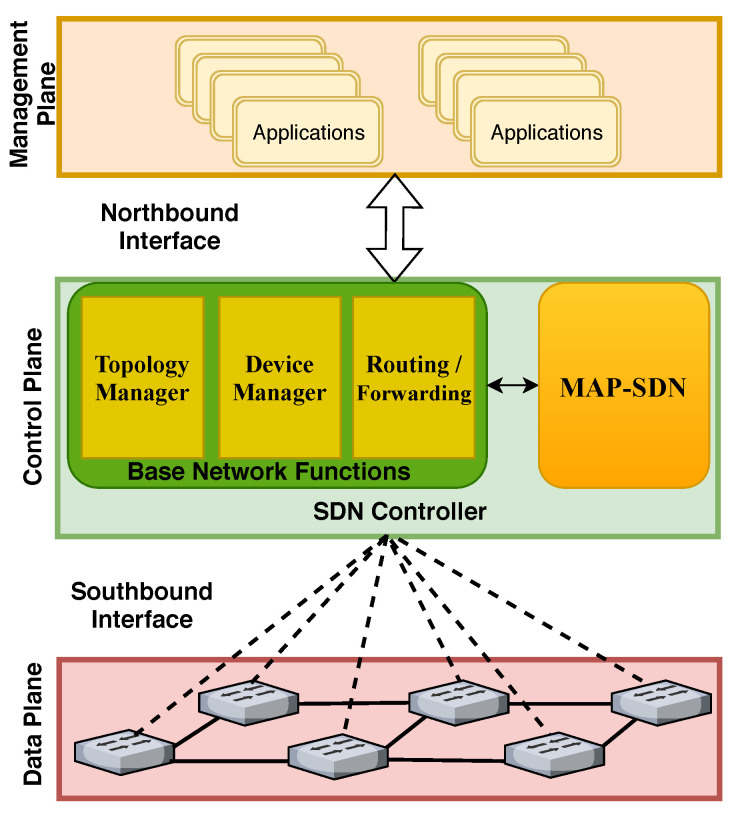
Layered view of SDN Architecture.

**Figure 2 sensors-22-08434-f002:**
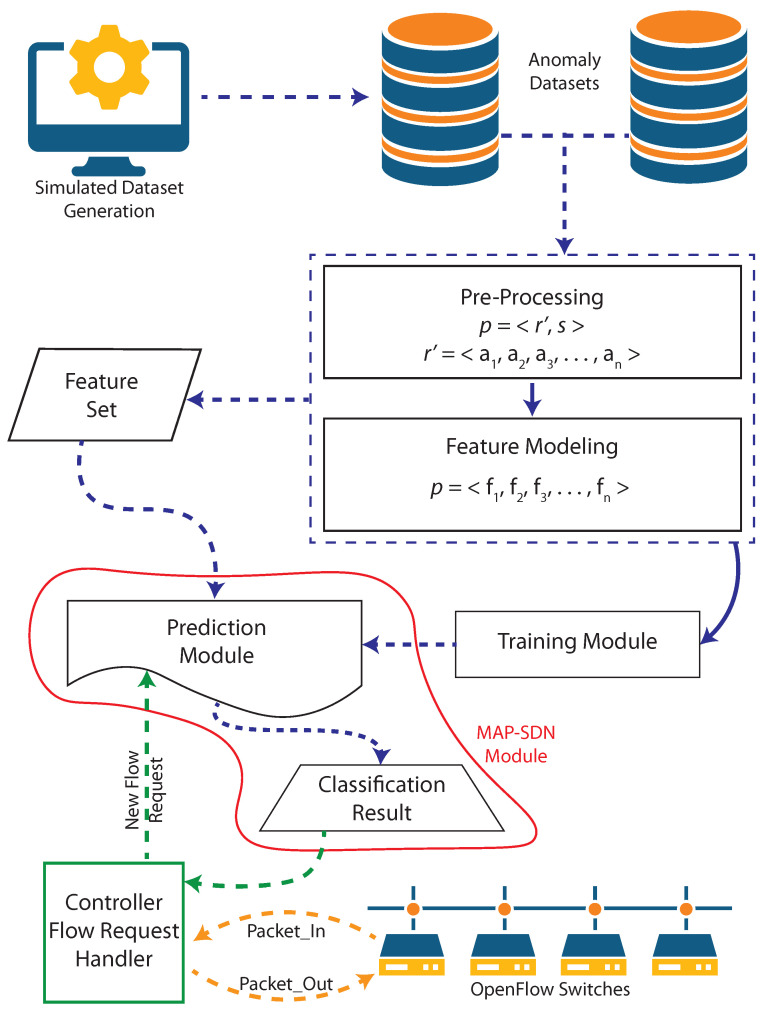
Overview of the proposed approach where dashed lines represent data to/from processes, and solid lines represent process-to-process communication.

**Figure 3 sensors-22-08434-f003:**
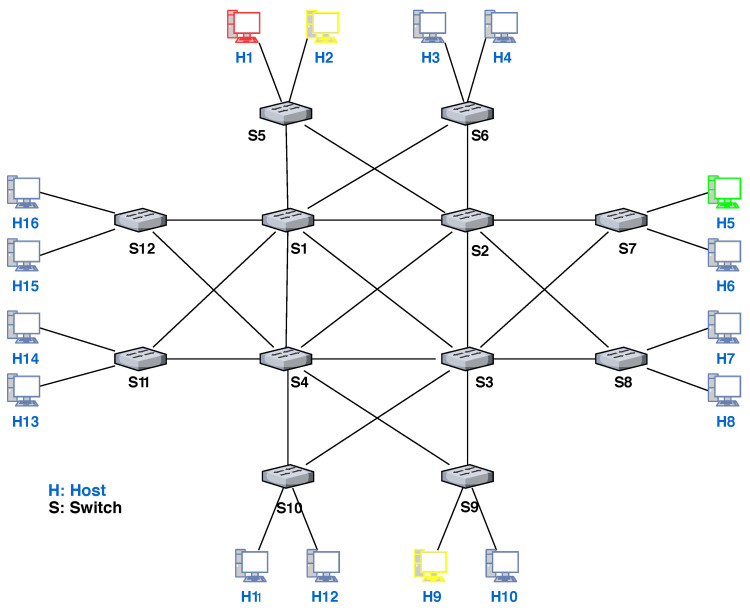
Topology used to generate an extensive synthetic dataset for training of classifier. Core: S1–S4, Edge: S5–S12, and Hosts: H1–H16.

**Figure 4 sensors-22-08434-f004:**
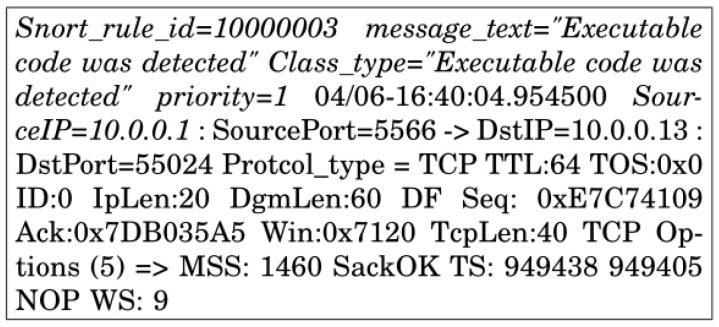
The output of Snort as a structured file.

**Figure 5 sensors-22-08434-f005:**
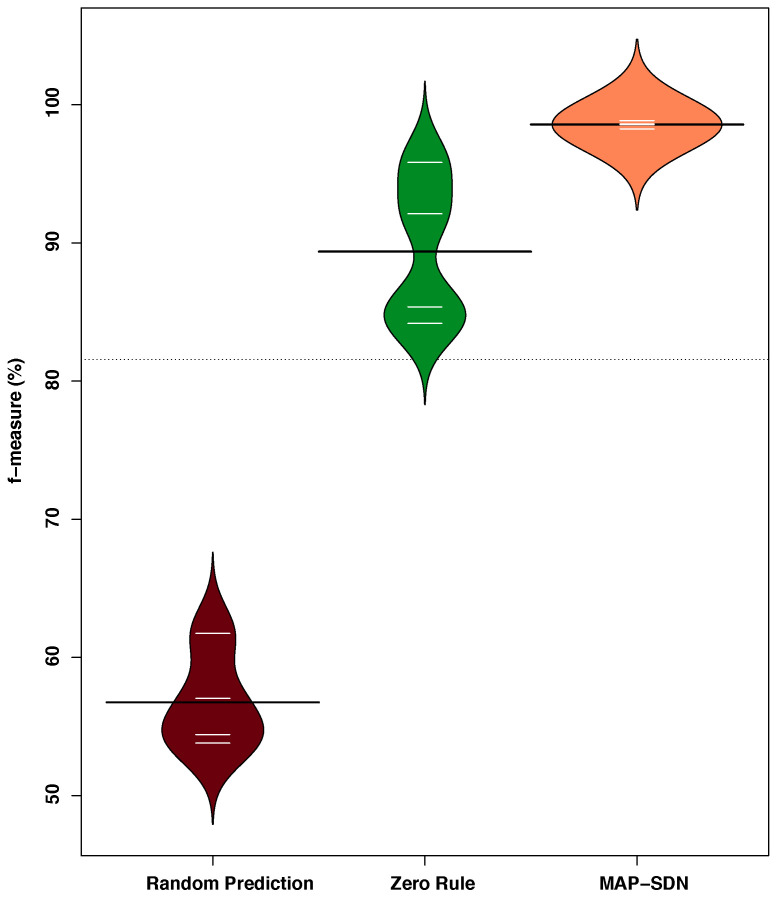
Accuracy distribution of MAP-SDN compared to RPA and ZRA.

**Figure 6 sensors-22-08434-f006:**
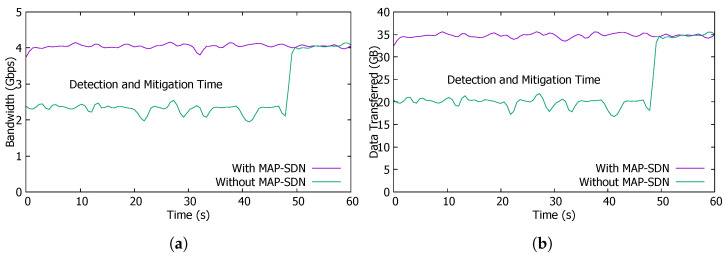
Real-time Performance Comparison of MAP-SDN. (**a**) Bandwidth with/without anomaly traffic. (**b**) Data transfer with/wihtout anomaly traffic.

**Table 1 sensors-22-08434-t001:** Major acronyms used in the paper.

Acronym	Description
API	Application Programmable Interface
BN	BayesNet
DDoS	Distributed Denial of Service
DT	Decision Tree
GNN	Graph Neural Network
GRU-RNN	Gated Recurrent Unit Recurrent Neural Network
GRU-LSTM	Gated Recurrent Unit Long Short Term Memory
HAA	Hierarchical Adversarial Attack
HMM	Hidden Markov Model
ICMP	Internet Control Message Protocol
IP	Internet Protocol
KNN	K-Nearest Neighbor
LR	Linear Regression
MAP-SDN	Machine Learning-Based Anomaly Prediction in SDN
MNB	Multinomial Naive Bayes
NB	Naive Bayes
NIDS	Network Intrusion Detection Systems
PCAP	Packet Capture
RF	Random Forest
RPA	Random Prediction Algorithm
SDN	Software Defined Networking
SVM	Support Vector Machine
TCP	Transmission Control Protocol
UDP	User Datagram Protocol
ZRA	Zero Rule Algorithm

**Table 2 sensors-22-08434-t002:** Comparison against baseline approaches.

	Proposed Approach				RPA				ZRA			
**Testing Samples**	**Accuracy**	**Precision**	**Recall**	**F-Measure**	**Accuracy**	**Precision**	**Recall**	**F-Measure**	**Accuracy**	**Precision**	**Recall**	**F-Measure**
Latest 5%	95.29%	99.26%	98.46%	98.86%	60.00%	57.49%	51.65%	54.42%	88.58%	84.92%	85.84%	85.37%
Latest 10%	95.34%	98.83%	98.39%	98.61%	61.00%	67.89%	56.62%	61.74%	91.82%	82.58%	85.84%	84.18%
Latest 15%	95.21%	98.55%	98.57%	98.56%	61.97%	53.29%	54.32%	53.80%	93.44%	95.93%	95.73%	95.83%
Latest 20%	95.25%	98.16%	98.36%	98.26%	64.25%	60.18%	54.22%	57.04%	90.46%	91.84%	92.38%	92.11%
Average	95.27%	98.70%	98.45%	98.57%	61.81%	59.71%	54.20%	56.75%	91.07%	88.82%	89.95%	89.37%

**Table 3 sensors-22-08434-t003:** Comparison against off-the-shelf classifiers.

Approach	Accuracy	Precision	Recall	F-Measure
RF	95.27%	98.70%	98.45%	98.57%
MNB	94.49%	94.53%	99.30%	96.85%
LR	93.12%	93.96%	98.27%	96.06%
SVM	90.27%	91.75%	84.46%	87.94%

**Table 4 sensors-22-08434-t004:** Influence of preprocessing.

Preprocessing	Accuracy	Precision	Recall	F-Measure
Enabled	95.27%	98.70%	98.45%	98.57%
Disabled	93.34%	95.64%	94.70%	95.17%
Improvement	2.07%	3.20%	3.95%	3.57%

## Data Availability

Not applicable.

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
