# Peer review of "A Machine Learning-Based Anomaly Prediction Service for Software-Defined Networks"

_sensors, 2022, doi:10.3390/s22218434_

Round 1
Reviewer 1 Report
- This paper talks about the Software Defined Networking (SDN), which has gained tremendous growth and can be exploited in different network scenarios from data centers to wide-area 5G networks. It shifts control logic from the devices to a centralized entity (programmable controller) for efficient traffic monitoring and flow management. A software-based controller enforces rules and policies on the requests sent by forwarding elements but cannot detect anomalous patterns in the network traffic. These anomalies may indicate threats to the network and decrease its performance and security. Machine learning (ML) approaches can identify such patterns in traffic flow and predict the impending threats to the systems. In this work, we propose an ML-based service to predict traffic anomalies for software-defined networks. We first create a large dataset for network traffic by modeling a programmable data center with a signature-based intrusion detection system. The feature vectors are pre-processed and are constructed against each flow request by the forwarding element. Then, we input the feature vector of each request to a machine learning classifier for training to predict anomalies. Finally, we use the hold-out cross-validation technique to evaluate the proposed approach. The evaluation results specify that the proposed approach is highly accurate and its average accuracy, precision, recall, and f-measure are 95.27%, 98.70%, 98.45%, and 98.57%, respectively. - This paper has potential. However, it requires a revision. - The abstract is well-written. However, it lacks a clear problem statement. - There must be a nomenclature added after the keywords. All the acronyms and symbols used in the paper must be added in the alphabetical order so that the reader than easily locate the relevant abbreviations. - What is the contribution of the paper? It must be clearly mentioned in the introduction. So far, this element is there but needs more clarification. - What motivated you to write this paper? It must be clearly mentioned in the introduction. So far, this element is not obvious in the introduction. - What is the scope and possible potential of the paper? It must be clearly mentioned in the introduction. So far, this element is not obvious in the introduction. - There should be a pseudo code to convolute Section IV. Pseudo codes are designed for practitioners to understand things smoothly. The variables of input and output are required to be defined in the start of the pseudo code. And then that defined language of variables are required to be used in the rest of the lines of each algorithm. - In section V of implementation and results, what main-stream techniques are utilized for comparison? The superiority of your proposed scheme shall be mentioned in the conclusion based on the outcome of comparison with other main-stream techniques. - For a Journal paper talking about anomaly prediction and detection in software defined networks, a more broader spectrum must be given for relevant industries. For example, industry of energy and power are also impacted from these anomalies in the form of cyber-attacks. A discussion on the same must be flavored more in the introduction. Similarly, the discussion on cyber-attacks will also maintain the balance of the work. Some suggested references on the same can be considered to added. - `Learning‐Based Methods for Cyber Attacks Detection in IoT Systems: A Survey on Methods, Analysis, and Future Prospects’, MDPI —Electronics, vol. 11(9), pp. 1–20, May 2022. - The figure quality is required to be improved. The text inside figures should be visible. - References literally require an overall. Most of the references are conference papers and technical reports. Are they relevant to the work ? Moreover, the format is not consistent. In some references, there is no page number mentioned. In others, the page number is at the end and year in the middle and vice versa. Also, in some, the first letter of the title of the paper is only capital. In others, all first letters of the titles of the paper are capital. In some references, the et al. is used to mention the list of authors. In others, the full list of co-authors is there. Please visit the styling again and maintain consistency.
Author Response
Dear Editors and Reviewers:
Thank you for your careful review and constructive comments regarding our manuscript "A Machine Learning based Anomaly Prediction Service for Software De ned Networks" having Manuscript ID: (Sensors-1972794). The comments have been very helpful in improving our work. The changes made in light of the comments are marked red in color in the updated version of the manuscript.
A list of major modifications includes:
In light of the comments, all suggested changes have been made. The major
shortcomings are addressed, and text has been improved.
We take this opportunity to express our appreciation for your expertise and invaluable assistance in reviewing our draft. We hope that the revised draft provides you a better presentation of our work. Should you have any questions, please do not hesitate to contact us.
Respectfully yours,
Zohaib Latif, Choonhwa Lee, Hanyang University, Seoul, Republic of South Korea
Qasim Umer, COMSATS University Islamabad, Vehari Campus, Pakistan
Kashif Sharif, Fan Li, Beijing Institute of Technology, Beijing, China
Sujit Biswas, University of East London, London, UK

Reviewer 2 Report
The paper focuses on intrusion detection in SDN-enabled networks. It is well-structured, has a good list of references, figures and tables are of good quality as well. However, I have many questions regarding the approach used...
First of all, terminology. Anomaly detection is usually referred to unsupervised machine learning, i.e. a model is trained using normal samples to detect those ones that deviate from the established norms. In this study, straightforward classification models are used, this is more like "intrusion detection", "attack detection", "traffic classification", etc. It can confuse a reader.
Second, if I understood correctly (and inability to understand correctly is also an issue in this manuscript), there is no connection between the traffic used for the analysis and SDN, i.e. the dataset used in the study is basically generated from the headers of IP packets, and therefore it is unclear what is the role of the SDN here. Similar effect could have been achieved by using a conventional switch and an IDS.
Key details of the approach are missing from the manuscript. For example, on page 6, there is no explanation how the traffic datasets were augmented with iperf an hping3. Did you just run those tools to generate some generic traffic that is obviously can be easily distinguished from the traffic related to attacks? Does not sound like a good idea to me. Could have used some of the publicly available datasets.
More details of the deployment of the MAP-DSN on the Floodlight controller should be provided in Section 4.
Was Snort used literally just for extracting features from IP headers? Is not it an overkill? Would not it better (faster) to use directly libpcap?
Concerning the ML algorithm selected, 2007 called - they want their random forest back... I mean seriously, why not deep learning? At least add a neural network to the part where different algorithms are compared to each other. However, it may turn out that it will outperform RF here...
The numerical evaluation part should include some comparison to analogous approaches from other studies.
p. 5 lines 182 - 183: "In SDN, the first packet of a new flow trims a Packet_In message against which a flow is installed" - I am not sure this is correct, a citation is needed. To my knowledge, network engineers on contrary try to avoid this kind of scenario when the controller is learning network flows, instead SDN flows can / should be pre-installed on the switches to avoid the controller saturation with packet_ins: other (unknown) packets are just dropped in this case.
p. 5, lines 199 - 200: "most datasets have an uneven distribution of normal and abnormal traffic" - this does not look like a problem to me... should be explained why it is an issue.
Minor issues / typos:
p.2, line 70: "We first (Mininet)" you first what? :)
p. 5, line 177: should be something like "... traffic, whereas it avoids ..."
"data set" or "dataset"?
p. 6, line 209 - "an example" -> "An"
p. 8, line 254 - "questions, that need to be" -> "questions needed to be"
p. 8, line 255 - "Following this describe" - something is wrong here as well
p. 8, line 256 "Finally, we present the threat to validation for the proposed scheme" - what?
p. 8, line 274 - "Dataset an Metrics" - "and"?
Ok, I stopped checking for typos at this point, just check the language globally in the manuscript, there are tons of those little mistakes...
Author Response

(The authors gave the same response as above.)

Reviewer 3 Report
This study proposes an ML-based service to predict traffic anomalies for software-defined networks. Overall the proposed approach is effective. However, there are still some problems about this manuscript.
1. It is recommended to release the collected dataset and add it as a new contribution.
2. The proposed method is based on RF, which is a mature technique. The novelty is questionable. The authors should add more evidence for their contributions in this study over the existing studies which also use the RF model.
3. Only some ML classifiers are used as baselines. However, many DL models have already been used in the literature. The authors should add DL models as baselines.
4. The discussion of graph-based methods for IDS should be added, which is the new research direction in relevant fields. Refer to
Jiang, W. Graph-based deep learning for communication networks: A survey. Computer Communications 2022, 185, 40-54.
Zhou X, Liang W, Li W, et al. Hierarchical adversarial attacks against graph neural network based IoT network intrusion detection system[J]. IEEE Internet of Things Journal, 2022.
Lo W W, Layeghy S, Sarhan M, et al. E-GraphSAGE: A Graph Neural Network based Intrusion Detection System[C]. NOMS 2022-2022 IEEE/IFIP Network Operations and Management Symposium. IEEE, 2022: 1-9.
5. Some typos still exist. For example, in line 70, page 2:
"We first (Mininet) [11] and a signature-based intrusion detection system"
should be
"We first use Mininet [11] and a signature-based intrusion detection system"
Similar problems should be checked for the whole manuscript.
Author Response

(The authors gave the same response as above.)

Round 2
Reviewer 3 Report
Dear authors,
Thanks for revising and resubmitting the manuscript. Overall the manuscript has been improved a lot and I have no further comments.
Author Response
Dear Editors and Reviewers:
Thank you for your careful review and constructive comments regarding our manuscript ”A Machine Learning based Anomaly Prediction Service for Software Defined Networks” having Manuscript ID: (Sensors-1972794). The comments have been very helpful in improving our work. The changes made in light of the comments are marked red in color in the updated version of the manuscript.
A list of major modifications includes:
• In light of the comments, the quantitative performance is mentioned in the abstract.
• The discussion on the sensor networks is added in the introduction section.
• Section III is updated in light of the comments.
We take this opportunity to express our appreciation for your expertise and invaluable assistance in reviewing our draft. We hope that the revised draft provides you with a better presentation of our work. Should you have any questions, please do not hesitate to contact us.
Respectfully yours,
All authors
